# Feasible Use of Cathode Ray Tube Glass (CRT) and Recycled Aggregates as Unbound and Cement-Treated Granular Materials for Road Sub-Bases

**DOI:** 10.3390/ma13030748

**Published:** 2020-02-06

**Authors:** M. Cabrera, P. Pérez, J. Rosales, F. Agrela

**Affiliations:** Area of Construction Engineering, University of Cordoba, 14007 Cordoba, Spain; manuel.cabrera@uco.es (M.C.); ir2pegop@uco.es (P.P.); jrosales@uco.es (J.R.)

**Keywords:** cathode ray tube glass, recycled aggregates, civil infrastructures, recycled aggregates, cement-treated materials

## Abstract

In the last 15 years, new types of display technologies have increasingly replaced cathode ray tube (CRT) screens, which has led to an increase in landfill of old discarded CRT televisions, which present a great environmental challenge throughout the world due to their high lead content. In addition, environmental awareness has led to greater use of recycled aggregates to reduce the exploitation of existing reserves. This document aims to study the feasibility of incorporating CRT glass waste with recycled aggregate (RA) in combinations for use in civil engineering, more specifically in road bases and sub-bases. For the mechanical and environmental assessment of all of the samples and materials, the following procedures have been performed: the compliance batch test of UNE-EN 12457-4:2004 for RA, CRT, and mixtures; the Percolation Test according CEN/TS 14405 for the mixtures, CRT, and RA; Modified Proctor and load capacity (the California Bearing Ratio, or CBR) in all mixtures without cement addition, and finally, compressive strength of the material treated with cement at different ages of curing. The analysis of the mechanical and environmental properties through different techniques of lixiviation was positive, showing the ability to use CRT for certain dosage percentages mixed with recycled aggregates.

## 1. Introduction

The use of recycled aggregates in road construction has been studied in recent years, and this type of application should be a priority in the future [1]. It is possible to apply several types of recycled aggregates in low-intensity traffic roads (liR). A conventional liR is built applying a course layer of concrete or asphalt on the top of other layers, such as the base, sub-base, and subgrade.

Traditionally, natural materials such as crushed rock, selected gravel, and stabilised materials are used in the base and sub-base of the road. In the last decade, several studies have been carried out to investigate the possibility of using recycled concrete aggregates (RCA) [2,3,4] and mixed recycled aggregates (MRA) in bases and sub-bases to provide a viable option for the use of recycled materials from construction and demolition waste (CDW) [5].

The application of recycled aggregates in road construction can be done by means of bound or unbound layers [6,7]. Currently, mixed recycled aggregates (MRA), which include different constituents of particles (concrete, ceramic, asphalt, masonry, natural, etc.), is the most common RA produced in different countries such as Spain, Portugal, Italy, etc. [8]. Some authors have studied the possibility of using these materials in applications with higher added value, such as the construction of unbound granular structural layers [5,9,10], and in cement-treated layers [11,12]. The main problem of the MRA is that they can have high amounts of sulphates, which cause dimensional changes due to the formation of ettringite [13].

Different authors have studied the mechanical behaviour of the RCA and MRA, and they demonstrated that cement-treated recycled materials presented an acceptable mechanical resistance up to 7 and 28 days when applying a 100% replacement rate of conventional materials (soil or gravel) by MRA or RCA [7].

Real-scale trials in which RCA and MRA were applied in cement-treated applications in roads have been conducted [12,14] in Malaga, Spain. In these studies, it was demonstrated that the behaviour of sections where RCA and MRA treated with cement were applied presented similar properties as those where conventional materials were used. In the work of Agrela et al. [12], it was observed that by applying a 100% replacement of natural aggregate (NA) by MRA with 3% of cement, an appropriate mechanical and deformational behaviour was achieved in the long term.

Another industrial waste that could be used in the sub-base of road layers are the residues from cathode ray tube televisions and screens (CRT waste). Although these types of residues are decreasing every year, there are still thousands of tons of CRT equipment still to be recycled. It is estimated that around the world, only about 26% of CRT waste is recycled, and the remaining 59% is deposited in landfills without recovery [15].

The main composition of CRT waste media consists of barium, strontium, and lead silicate [16,17]. These waste are also rich in silica, which makes their use as a building material interesting; however, other experiences demonstrate that too high content of silicates can degrade the mechanical strength [18]. Different investigations have been carried out for the second life cycle of CRT waste in the field of construction. Hui and Sun [19] studied the properties of CRT waste as a replacement for the fine natural aggregate for the manufacture of mortars. Ling et al. [20] used CRT waste cement mortar for X-ray radiation-shielding applications. Romero et al. [21] studied the mechanical properties of concrete manufactured with different CRT waste protocols as a substitute for natural aggregates, and Abdallah and Fan [22] studied the characteristics of concrete with waste glass.

The main problem of CRT waste is its lead content, and its contamination potential has been studied by different authors. Ling and Poon [23] observed that the back of the CRT equipment had a higher lead content than the front, so its use in construction applications could cause problems with respect to the environment and can cause public health problems. However, recycled aggregate (RA), particularly MRA, the sulphate content of the recycled aggregates (mostly gypsum), is one of the important quality properties for recycling and classification.

This aims of this study is to investigate the possibilities of applying MRA with CRT waste in a combined manner, including a reduced percentage of CRT to avoid the possibility of the leaching of contaminant elements. To this end, several studies will be carried out on mixtures of MRA and CRT waste in different proportions, studying both their mechanical behaviour and the potential contamination that they could present. It is applied to different mixtures using mainly the CRT waste coming from the front of the equipment with very small quantities from the back of the equipment, which could be more polluting.

## 2. Materials

Cement: The cement used in this study was CEM II/B-L 32.5 N (referred to as CEM-II). The main properties of the cement are shown in Table 1. This cement is the most commonly used material in this type of application due to its medium hydration heat and high resistance to chemical attack.

Mixed Recycled Aggregates (MRA): These came from the CDW treatment of the Aristerra S.L. plant, located in Malaga (Spain). The composition of the MRA, determined according to the standards UNE-EN 13242 [24] and UNE-EN 933-11 [25], must exceed 70% by weight in concrete, concrete products, mortars, pieces for the manufacture of concrete masonry, aggregates, and natural stones, as well as materials treated with hydraulic binders. It cannot exceed 2% by weight of glass. The rest consisted of ceramic masonry materials made of clay (bricks and tiles) or calcium silicate.

Recycled asphalt pavement (RAP): This material comes from the treatment and crushing of asphalt pavements, and it is usually used in the construction of agglomerate layers of new roads. Sometimes, they are waste that arrives at the RCD treatment plants, and they are mixed to obtain MRA. This material also comes from the plant Aristerra S.L.

Cathode ray tube glass (CRT): The material coming from the recycling of electronic devices was received in the construction engineering laboratory of the University of Córdoba from landfill. The blocks of this waste were large, so it was decided to crush it prior to testing. The CRT is divided in two types. One side is coming from the front part where the image is displayed, called p-CRT_F_ (processed front CRT), and on the other side is obtained from the rear part composed by the hood or cathode ray tube, and it is called p-CRT_R_ (processed rear CRT). Figure 1 shows both parts from which come these CRT types.

MRAa: A mixture of MRA and processed RAP was prepared in the laboratory by mixing 75% MRA and 25% RAP in dry weight. The mixture was called MRAa.

Figure 2 shows the particle size distribution of the materials. It was observed that the RAP contains a particle size distribution with a lower percentage of fine particles compared to the rest of the materials analysed. This is due to its crushing process. The rest of the materials presented a continuous particle size distribution.

In Table 2, the data of the physical and chemical properties of the materials tested in the laboratory are shown. The MRA presented a lower density of saturated surface and a greater absorption of water due to the content of ceramic particles [12,26]. The coarse fraction of the p-CRT presents a near-zero absorption. The sulphate content in all materials does not exceed the Spanish regulations. CRTs are made with two different glass formulations, one for the front, p-CRT_F_ (panel) and one for the rear, p-CRT_R_ (funnel). p-CRT_R_ contains lead silicate glass with a composition of SiO_2_ 51.20 wt%, PbO 23.14 wt%, and other oxides 25.66 wt% p-CRT_F_ contains barium–strontium silicate glass with SiO_2_ 58.13 wt%, BaO 10.50 wt%, SrO 9.46 wt%, and other oxides 21.91 wt%.

A classification test for the constituents of coarse recycled aggregates was carried out in accordance with the UNE-EN 933-11: 2009 [25] standard. Manual separation of the recycled aggregate components was carried out on particles of over 4 mm in size to obtain the results shown in Table 3.

X-ray diffractometry was used to study both types of p-CRT. Figure 3 shows the XRD pattern of the samples analysed. The pattern obtained in both materials shows a mostly crystalline material. The p-CRT_F_ shows mainly high contents of Cd, Mg, Si, and Ni; however, the p-CRT_R_ is mainly formed by Pb and Si, which are elements present in both preliminary raw materials.

### Mixtures of Recycled Materials and p-CRT Studied

Different mixtures were made to study the possibility of use as a granular material and as cement-treated granular material (CTGM).

The p-CRT used for the study was a combination of the front and back. The proportion used was 2/3 p-CRT_F_ and 1/3 p-CRT_R_. The percentage of p-CRT that was added to the mixtures was 10% dry mass. Table 4 shows the mixtures and proportions used in the study.

## 3. Classification of Materials as a Function of their Pollutant Potential

### 3.1. Compliance Test UNE EN 12457-4:2004 [34]

Compliance testing was conducted to check whether the five materials satisfied European regulations. To classify these materials according to the EU Landfill Directive, not only heavy metals, but also inorganic anions were measured (sulphate, chloride, and fluoride). The UNE-EN 12457-4 [34] procedure consists of a two-step batch leaching test and is performed on materials with particle-size dimensions below 10 mm. The solid/liquid ratio is 1:10. The batch reactor is continuously stirred (10–12 rpm) for 24 h at a controlled temperature value equal to 20 ± 5 °C. At the end of the 24 h, the samples are left to decant and the pH, conductivity, and temperature are measured. The solution is filtered using a membrane filter (0.45 μm), and a sub-sample of the leachate was taken for each material.

Directive 2033/33/CE establishes the limits to be admitted to landfill in three categories (inert, non-hazardous, and hazardous waste) based on the concentration of heavy metals obtained in the UNE 12457-4 [34] compliance test. The measured concentrations in the leachate (mg/kg) are shown in Table 5 (data in bold indicate that the value exceeds the limit for inert waste).

Comparing the measured concentrations of species released with the European environmental criteria at liquid-to-solid ratio of 10 L/kg (see Table 5), the calculated admission values for the MRA and p-CRT showed some critical parameters for their use: Cr and Pb.

The MRA was classified as non-hazardous waste due to the fact that the limit of inert waste was exceeded for Cr, according to other studies [35,36]. The presence of Cr was due to the content of ceramic particles.

The p-CRT contains 20–25% of PbO and is classified as hazardous waste [37]. Its content of Pb in leaching is a very important factor for its application as a construction material because it can represent a threat to the environment and human health. The tested sample of p-CRT_R_ is classified as a non-hazardous waste. The mixture of MRA with p-CRT is classified as an inert material, its use being adapted from the environmental point of view.

### 3.2. Percolation Test CEN/TS 14405 [38]

The Column Leach Test describes a procedure for determining the leachability of inorganic components from solid, earthy, and stony materials and wastes as a function of the value of L/S. The method involves passing demineralised water upwards through a vertical column of particulate material (4 mm or smaller). Seven consecutive leachate fractions are collected, corresponding to a liquid-to-solid ratio range of 0.1 L/kg (v/m).

After the basic characterisation of the release of pollutants by the compliance test by which the most conflicting elements are identified, the percolation test was carried out according to the procedure CEN/TS 14405 [38] to evaluate the release of components with a liquid/solid ratio = 0.1 [39].

Table 6 shows the leachate concentrations obtained according to the percolation test. It is observed that mixtures with p-CRT behave as an inert material, corroborating the data obtained in the compliance test. Only p-CRT_R_ exceeds the limits of an inert material.

Theoretically, the Pb results of the percolation leaching test of MRA + p-CRT samples should be 0.0118 mg/L, but the results have a much lower value, which suggests that recycled aggregates absorb part of the contaminating power of p-CRT glass.

## 4. Experimental Methods and Results

### 4.1. Modified Proctor (UNE 103501:1994) [40]

The Modified Proctor test is a laboratory geotechnical testing method used to determine the soil compaction properties, specifically, to determine the optimal water content at which soil can reach its maximum dry density.

Modified Proctor has a real importance in the construction industry. The structures need a resistant base to lean on, and a poorly compacted soil could mean the collapse of a well-designed structure. In some cases, such as on roads with little traffic or rural areas, the soil constitutes the rolling layer, so the importance of compaction becomes evident.

As shown in Figure 4, all materials have curves that are not very sensitive to changes in moisture content. Arulrajah et al. [41] reported that the materials with flat compaction curves can tolerate a greater amount of variation in moisture content without compromising much of the density obtained from compaction.

The material with the highest humidity and the lowest density is the MRA (Table 7), due to its content of ceramic particles. This content of particles increases the porosity and therefore produces a lower density of the material and a greater absorption. The mixtures of MRA and MRAa with p-CRT increase the maximum dry density.

### 4.2. California Bearing Ratio (CBR)

The California Bearing Ratio (CBR) test is a strength test that compares the bearing capacity of a material with that of a well-graded crushed stone (thus, a high-quality crushed stone material should have a CBR of 100%). It is primarily intended for, but not limited to, evaluating the strength of cohesive materials having maximum particle sizes less than 20 mm according to UNE 103502-95 [42].

The CBR test involves comparing the application of one ratio of force per unit area required to penetrate a soil mass with a standard circular piston at the rate of 1.25 mm/min to that required for the corresponding penetration of a standard material.

Das [43] said that the bearing capacity of the soil is the capacity that the material has to support loads applied on it, without the failure that occurs by cutting or a large deformation.

The loads that a structural layer of a road transmits to the ground produce tensions and deformations on it. The deformation will depend on the tension and the physical–mechanical properties of the terrain. In pavements, the transmitted load is mobile, experiencing cycles of loading and unloading, where a part of the deformed land is recovered, and another part is not. It should be noted that the CBR test indicates the quality of support that a soil will have. Authors such as Crespo [44] classify the soil based only on the CBR index.

The CBR test was performed on the compacted samples at the optimum moisture content, for which the Modified Proctor compaction test was used. CBR tests were carried out in both unsoaked and 4-day soaked conditions, and the results are summarised in Figure 5.

According to other studies [45], in an unsoaked condition, the materials showed the highest CBR value. The CBR value decreased with the p-CRT content in both conditions. One possible reason was the lower intrinsic resistance to the crushed p-CRT particles.

### 4.3. Vibrating Hammer Times

The materials were compacted in a CBR mould using a vibrating hammer in accordance with NLT-310/90 [46]. They were filled in three layers, each layer formed with a thickness of one-third of the length of the mould to produce the density/time plot. The compaction value required for each sample must be greater than 98%, according to the modified Proctor reference value. Three samples were manufactured using different vibration times of 5, 12, and 20 s.

The compaction value required for each specimen must be greater than 98%, according to the modified Proctor reference value. The compaction times of the vibrating hammer were different in all the mixtures. Materials not mixed with p-CRT showed high compaction times. Figure 6 shows the time required to obtain a vibration hammer of 98% Proctor density. Vibrating hammer compaction times range from 17 to 19 s.

### 4.4. Compressive Strength

The purpose of this test is the determination of the simple compressive strength of the materials treated with hydraulic binders. This test has been carried out according the methodology indicated by NLT-305/90 [46].

The simple compressive strength in mixtures treated with cement is an indicator of the degree of reaction of the soil with cement and water and the rate of hardening with respect to time. The values obtained depend on many factors: the content and type of cement, type of soil, the applied energy of compaction, the efficiency achieved in the mixing, amount of organic matter, size and shape of the test specimen, etc. Particularly, due to the morphology and low water absorption shown by the CRT, the interaction between the particles and the cement does not materialize as with a natural or recycled aggregate, obtaining a loss of resistance in the mixtures that incorporate CRT.

In the case of pavement structures, there are compressive strength values suggested according to the type of pavement and type of layer to be built. In Spain, the regulation requires a minimum resistance of 2.5 MPa to 7 days, with a cement percentage not lower than 3%.

Figure 7 shows the results obtained in all the CTGM mixtures produced at 7, 28, and 90 days. Dashed lines indicating the minimum compressive strength that must be obtained after 7 days (2.5 MPa), as well as the maximum resistance to 28 days (4.5 MPa) in accordance with the Spanish technical specifications, are also shown.

It is observed that all the mixtures exceed the required limit at 7 days. The highest values of compression resistance were obtained in the MRA mixtures, with similar results obtained by other authors [13].

Figure 8 shows the evolution of the strength. At 28 days, compression strength values of approximately 90% were obtained in comparison with the resistance values obtained at 90 days. At 7 days, the results were approximately 80% compared to the results measured at 90 days. The addition of p-CRT decreases the resistance in both mixtures. If we compare the MRA with MRA + p-CRT and MRAa with MRAa+ p-CRT, there is a decrease in resistance at 28 days of 12.97% and 14.58%, respectively. The compressive strength of all the mixtures exceeds the minimum value of 2.5 MPa, corroborating the use of p-CRT in mixtures treated with cement.

## 5. Conclusions

This paper provides the results of research on cathode ray tube glass in combination with mixed recycled aggregates (p-CRT and MRA) with different compositions to be used as sub-bases and bases of roads pavements, obtaining the following conclusions:Regarding the physical and chemical properties of the materials, the mixed recycled aggregates presented a lower density and greater absorption of water due to its content of ceramic particles (21.16% of ceramic particles). Los Angeles abrasion testing indicates a good quality of all aggregates in accordance with current regulations (>40).Potential contamination by leaching was studied by means of two different methods, compliance testing (UNE-12457-4) [34] and percolation testing (CEN/TS-14405) [38]. The compliance test data revealed a potential contaminant in the funnel (p-CRT_R_) of the analysed samples, due to the high content of Pb, confirming that they are an inappropriate material for the application as an isolated aggregate in any civil engineering application.

The present study indirectly demonstrates the neutralised capacity of mixed recycled aggregates from an environmental point of view, reducing the concentration of heavy metals when mixed with cathode ray tube glass (see Table 5).

The results show that the use of cathode ray tube glass as an aggregate, at 10%, not only had satisfactory levels of compressive strength and bearing capacity, but also does not exceed the limits established by the directive, all mixtures being classified as inert.

The viability of using cathode ray tube glass and mixed recycled aggregates as an aggregate for the production of structural road layers (base and sub-base) has been demonstrated.

Therefore, the present work proposes a solution that implies an environmental benefit for these agents in addition to verifying the samples (as unbound aggregates and as cement-treated granular materials mixtures) with potential to be applied as building materials during their second life cycle.

## Figures and Tables

**Figure 1 materials-13-00748-f001:**
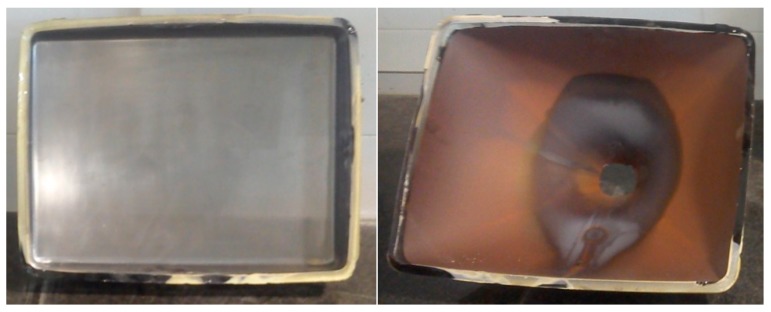
**Left**, front part cathode ray tube (CRT_F_); **right**, rear part (CRT_R_).

**Figure 2 materials-13-00748-f002:**
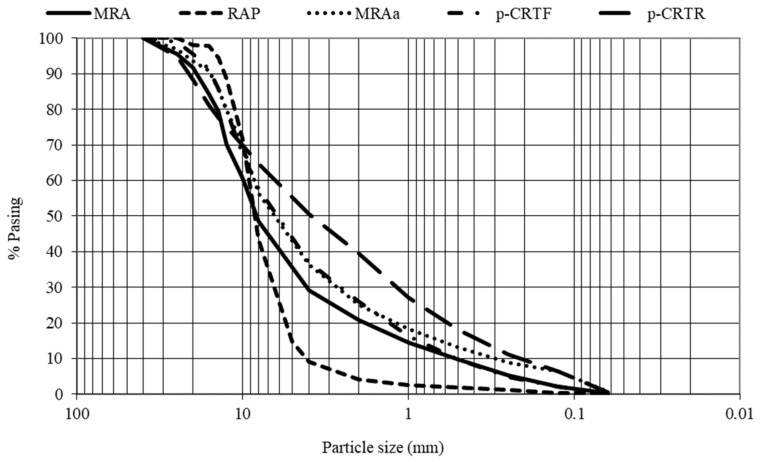
Particle size distribution.

**Figure 3 materials-13-00748-f003:**
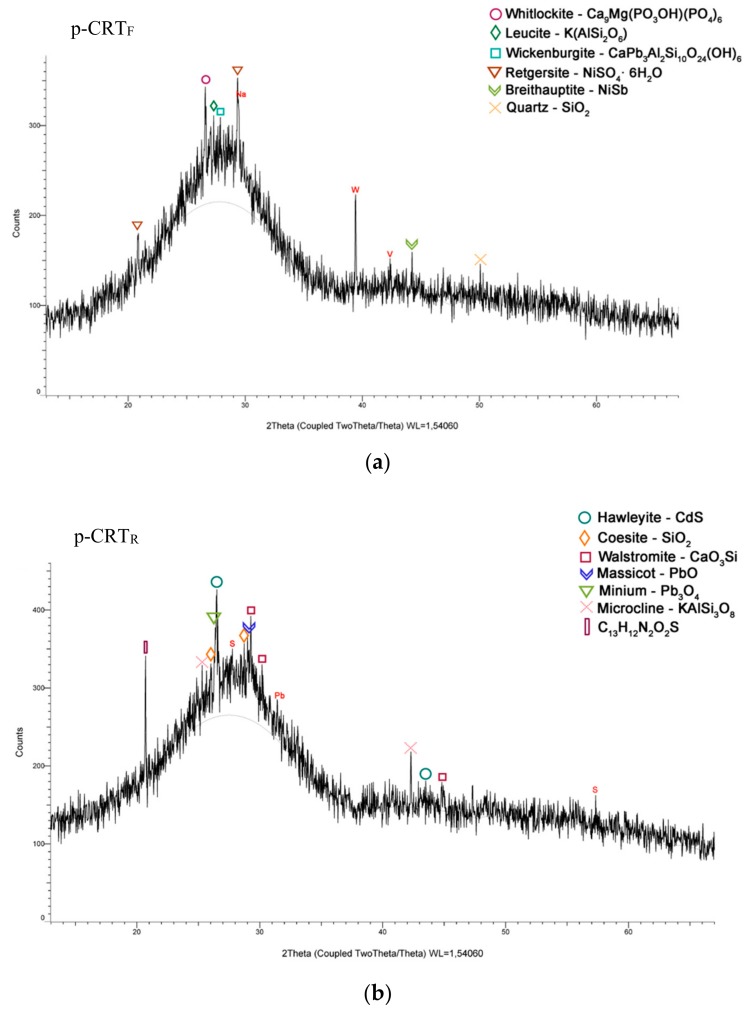
(**a**) X-ray diffractogram of p-CRT_F_; (**b**) X-ray diffractogram of p-CRT_R_.

**Figure 4 materials-13-00748-f004:**
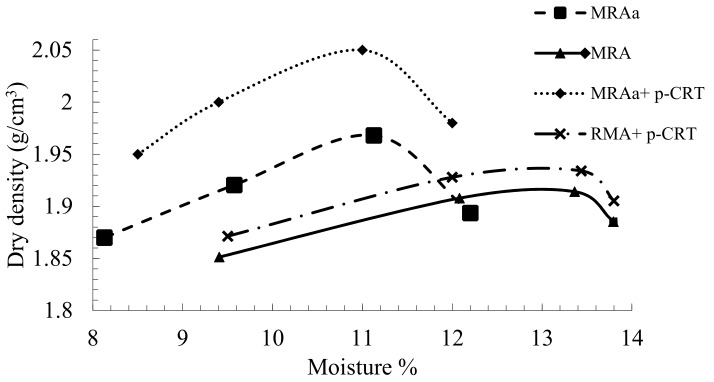
Moisture–density relationship.

**Figure 5 materials-13-00748-f005:**
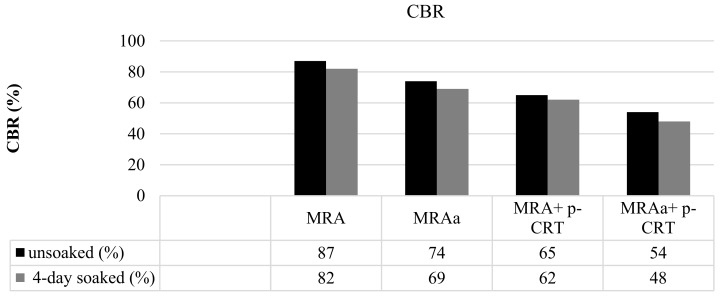
California Bearing Ratio (CBR) values (unsoaked and 4-day soaked).

**Figure 6 materials-13-00748-f006:**
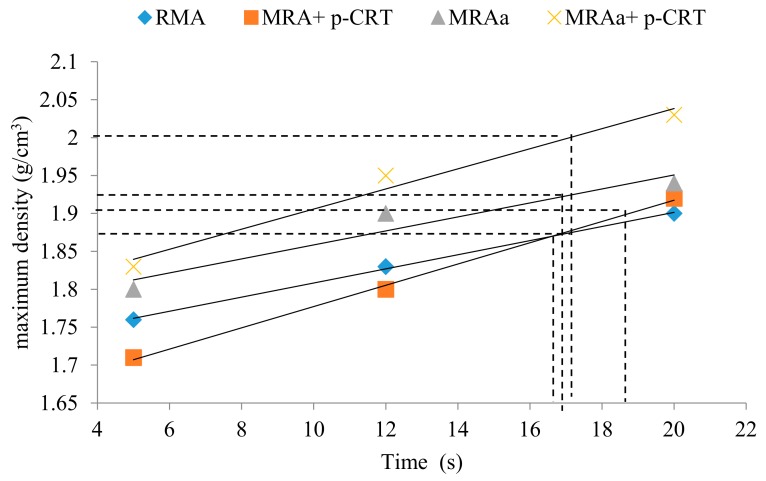
Vibrating hammer times.

**Figure 7 materials-13-00748-f007:**
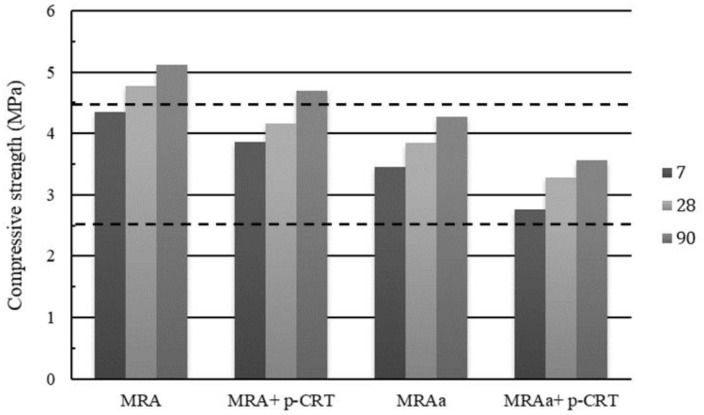
Compressive strength results with 3% cement.

**Figure 8 materials-13-00748-f008:**
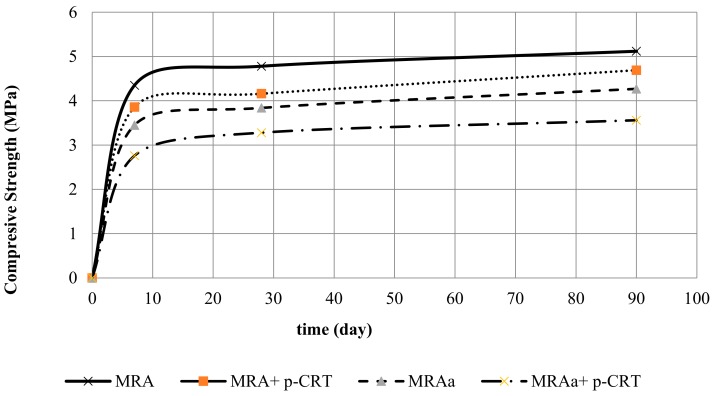
Compressive strength evolution.

**Table 1 materials-13-00748-t001:** Properties of the cement.

Cement	SiO_2_	Al_2_O_3_	Fe_2_O_3_	CaO	MgO	SO_3_	K_2_O	Loss on Ignition (975 °C)
CEM II (%)	26.24	8.7	3.36	54.06	1.34	3.32	1.44	1.3

**Table 2 materials-13-00748-t002:** Physical and chemical properties of recycled aggregates and glass waste electrical. MRA: mixed recycled aggregates, RAP: recycled asphalt pavement, MRAa: mixture of 75% MRA and 25% processed RAP, p-CRT_R_: cathode ray tube with lead silicate glass and a composition of SiO_2_ 51.20 wt%, PbO 23.14 wt%, and other oxides 25.66 wt%. p-CRT_F_: cathode ray tube with barium–strontium silicate glass and a composition of SiO_2_ 58.13 wt%, BaO 10.50 wt%, SrO 9.46 wt%, and other oxides 21.91 wt%.

Properties	MRA	RAP	MRAa	p-CRT_R_	p-CRT_F_	Test Method
Acid-soluble sulphate (%SO_3_)	0.7	0.3	0.39	0.05	0.03	UNE-EN 1744-1 [27]
Organic material (%)	1.37	1.13	1.21	-	-	UNE 103204 [28]
Oxide content (%)	-	-	-	-	-	-
SiO_2_	53.12	-	50.75	51.2	58.13	UNE 196-2 [29]
Al_2_O_3_	13.15	-	10.6	4.15	2.71	-
TiO_2_	1.74	-	0.98	0.1	0.5	-
CaO	10.12	-	16.56	3.56	2.4	-
MgO	5.15	-	2.28	2.45	0.80	-
Na_2_O	2.8	-	1.97	7.3	8.10	-
K_2_O	1.7	-	2.31	8.1	7.30	-
Fe_2_O_3_	8.4	-	3.55	-	0.10	-
BaO	-	-	-	-	10.50	-
SrO	-	-	-	-	9.46	-
PbO	-	-	-	23.14	-	-
Other	<4%	-	<11%			-
Density (kg/m^3^)	-	-	-	-	-	UNE-EN 1097-6 [30]
0–4 mm	2.01	2.42	2.24	2.2	2.25	-
4–31.5 mm	2.08	2.26	2.12	2.49	2.51	-
Water absorption (%)	-	-	-	-	-	UNE-EN 1097-6 [30]
0–4 mm	10.27	5.1	9.41	6.27	5.76	-
4–31.5 mm	8.31	2.36	7.75	0.25	0.21	-
Plasticity	Non-plastic	Non-plastic	Non-plastic	Non-plastic	Non-plastic	UNE-EN-ISO 17892-12 [31]
Los Angeles	36	32	35	-	-	UNE-EN 1097-2 [32]
Friability ratio	27	21	24	31	33	UNE 146404 [33]

**Table 3 materials-13-00748-t003:** Composition.

	Concrete (Rc)	Natural Aggregates (Rc)	Ceramic (Rb)	Bituminous (Ra)	Glass (Rg)	Other (X)
MRA (%)	27.68	37.28	21.16	13.67	0.04	0.17
RAP (%)	-	-	-	100	-	-
MRAa (%)	22.61	31.31	19.73	26.72	0.03	0.11
p-CRT (%)	-	-	-	-	100	-

**Table 4 materials-13-00748-t004:** Dosages of the mixtures.

Mixtures	Materials (kg)		
	MRA	RAP	p-CRT
MRA	1000	-	-
MRAa	750	250	-
MRA + p-CRT	900	-	100
MRAa + p-CRT	675	225	100

**Table 5 materials-13-00748-t005:** Leachate concentrations (mg/kg) obtained by the compliance test UNE EN 12457-4.

Element	MRA	MRA + p-CRT	p-CRT_F_	p-CRT_R_	p-CRT	MRAa + p-CRT	Limit Values (mg/kg) Inert	Non Hazardous
	(mg/kg)	(mg/kg)	(mg/kg)	(mg/kg)	(mg/kg)	(mg/kg)		
Cr	0.52496	0.43146	0.03208	0.00524	0.02291	0.470579	<0.5	0.5–10
Ni	0.00817	0.010147	0.00635	0.02166	0.01133	0.016748	<0.4	0.4–10
Cu	0.06137	0.036868	0.01116	0.09846	0.04015	0.036969	<2	2–50
Zn	0.04186	0.053005	0.16044	0.40719	0.24241	0.031649	<4	4–50
As	0.01705	0.019116	0.00822	0.00518	0.00074	0.020214	<0.5	0.5–2
Se	0.00143	0.01233	0.00275	0.00491	0.00394	0.010007	<0.1	0.1–0.5
Mo	0.07182	0.032939	0.02103	0.00604	0.01519	0.036853	<0.5	0.5–10
Cd	0.00016	0.00001	0.00000	0.00000	0.00000	0.00000	<0.04	0.04–1
Sb	0.04046	0.041975	0.28803	0.24131	0.26973	0.042393	<0.06	0.06–0.7
Ba	0.16046	0.288945	16.4451	8.26438	13.6741	0.281764	<20	20–100
Hg	0.00018	0.000921	0.00001	0.00000	0.00000	0.001562	<0.01	0.01–0.2
Pb	0.00048	0.00374	0.48885	**2.47351**	**1.5493**	0.00001	<0.5	0.5–10
Cl^−^	21.5	50	300	40	197	18.5	800	15000
F^−^	<2	<2	<2	<2	<2	<2	10	150
SO_4_^−^	1150	1280	127	138	129	1240	1500	20000

**Table 6 materials-13-00748-t006:** Release obtained according the percolation test CEN/TS 14405.

Element	RMA	RMA	p-CRT_F_	p-CRT_R_	p-CRT	RMAa	Limit Values Directive 2003/33/EC (mg/L)	
		+				+		
		p-CRT				p-CRT		
	(mg/L)	(mg/L)	(mg/L)	(mg/L)	(mg/L)	(mg/L)	Inert	Non Hazardous
Cr	0.0052	0.0412	0.0087	0.0023	0.0065	0.0513	<0.1	0.1–2.5
Ni	0.0006	0.0052	0.0043	0.0020	0.0035	0.0057	<0.12	0.12–3
Cu	0.0010	0.0144	0.0219	0.0047	0.0160	0.0176	<0.6	0.6–30
Zn	0.0016	0.0079	0.0142	0.0160	0.0146	0.0097	<1.2	1.2–15
As	0.0006	0.0043	0.0030	0.0027	0.0028	0.0072	<0.06	0.06–0.3
Se	0.0016	0.0057	0.0011	0.0014	0.0012	0.0090	<0.04	0.04–0.2
Mo	0.0210	0.0811	0.0044	0.0038	0.0042	0.0820	<0.2	0.2–3.5
Cd	0.0001	0.0001	0.0000	0.0000	0.0000	0.0001	<0.02	0.02–0.3
Sb	0.0016	0.0077	0.0116	0.040	0.0208	0.0047	<0.1	0.1–0.15
Ba	0.0557	0.0615	0.2327	0.7725	0.4085	0.0972	<4	4–20
Hg	<0.0001	<0.0001	0.0001	0.0000	0.0001	0.0000	<0.002	0.002–0.03
Pb	0.0004	0.0018	0.0908	**0.1608**	0.1130	0.0028	<0.15	0.15–3

**Table 7 materials-13-00748-t007:** Dry density and optimum moisture results.

	MRA	MRA + p-CRT	MRAa	MRAa + p-CRT
Dry density (g/cm^3^)	1.92	1.94	1.97	2.05
Moisture (%)	13.29	13.19	11.1	11.01

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
