# Peer review of "Feasible Use of Cathode Ray Tube Glass (CRT) and Recycled Aggregates as Unbound and Cement-Treated Granular Materials for Road Sub-Bases"

_materials, 2020, doi:10.3390/ma13030748_

Round 1

Author Response

Reviewer#1: The work reports an experimental study for the evaluation of feasibility of mixing glass cathode ray tube waste with recycled aggregates for use in civil engineering, widening previous researches by the Authors.

The results are interesting. The paper is well written and the conclusions clear and promising. After carefully reading the manuscript the reviewer has the only following comments

In the introduction, line 40, some reference about other European experiences could be added (sse for instance DOI: 10.4401/ag‐7785 and related references).

In the introduction, line 61, the authors declare that the cathode ray tube waste are rich in silica,  which makes their use as a building material interesting. Other experiences demonstrate that too high content of silicates can degrade the mechanical strength.  Some comment could be added added    (see for instance  DOI:  10.1016/j.conbuildmat.2017.06.125). 

We completely agree with your suggestion, both works have been referenced in the paper

[4] Contrafatto, L., Cosenza, R., Barbagallo, R., & Ognibene, S. (2018). Use of recycled aggregates in road sub-base construction and concrete manufacturing. Annals of Geophysics, 61(2), 223.

[18] Contrafatto, L. (2017). Recycled Etna volcanic ash for cement, mortar and concrete manufacturing. Construction and Building Materials, 151, 704-713

In table 3 the units of measurement are missing, even if they are obvious. In table 4 RAB must be replaced with RAP 

It has been changed.

In section 3.4, line 255 it could be useful to explain the meaning of “maximum resistance at 28 day”, because mixtures MRAa, MRA+p-CRT and MRAa+p-CRT does not exceed that value at 28 days (figure 7).

According to the Spanish technical specifications, it is recommended not to have a resistance at 28 days greater than 4.5MPa.

Reviewer 2 Report

This manuscript is interesting and one of high quality and i suggeste to be published in your journal after minor revisions. More specifically:

line 50: you can replace the word "works" with the word "studies" 

line 73: you should rephrase the begining of the paragraph. My suggestion is : The aim of this study is to ......

line 121: In table 2 you should replace the Elemental Content (%) with Oxide content (%)

In the introduction part you could add a paragraph containing the use of different recycled materials which have been used by several authors as aggregates in concretes, such as 

Petrounias, P., Giannakopoulou, P.P., Rogkala, A., Lampropoulou, P., Tsikouras, B., Rigopoulos, I., Hatzipanagiotou, K. 2019. Petrographic and Mechanical Characteristics of Concrete Produced by Different Type of Recycled Materials. Geosciences, 9, 264. Abdallah, S., Fan. M. 2014. Characteristics of concrete with waste glass as fine aggregate replacement. International Journal of Engineering and Technical Research. 2, 11-17.

Best regards

Author Response

This manuscript is interesting and one of high quality and I suggest to be published in your journal after minor revisions.

More specifically:

line 50: you can replace the word "works" with the word "studies" 

It has been changed.

line 73: you should rephrase the begining of the paragraph. My suggestion is : The aim of this study is to ......

It has been changed.

line 121: In table 2 you should replace the Elemental Content (%) with Oxide content (%)

We are grateful by the opinion. It has been changed

In the introduction part you could add a paragraph containing the use of different recycled materials which have been used by several authors as aggregates in concretes, such as 

Thank you very much for your suggestion, Abdallah, S., & Fan, M. (2014) have been included in the document

[22] Abdallah, S., & Fan, M. (2014). Characteristics of concrete with waste glass as fine aggregate replacement. International Journal of Engineering and Technical Research (IJETR), 2(6), 11-17.

Reviewer 3 Report

The manuscript investigated the feasibility of using CRT glass waste with recycled aggregate as road subbases materials. The results showed the potential of using the mixed MRA and CRT for road subbases. The manuscript is more like a report rather than a research article. Section 3 described the results of the test, however, the analysis and discussion of the results are required:

In section 3.3, please discuss the reason that caused the difference of vibrating hammer times for different samples. In section 3.4, how does the CRT affected the mechanical properties of the designed mixtures? The reason for the change of the compressive strength with different mixtures needs to be discussed. Line 282-284, please discuss the possible reason for the neutralization effect of Mixed Recycled Aggregates in the article. In Table 3, what is RAA stand for? In Table 4, what are RMA and RAB stand for?

Author Response

The manuscript investigated the feasibility of using CRT glass waste with recycled aggregate as road subbases materials. The results showed the potential of using the mixed MRA and CRT for road subbases.

In section 3.3, please discuss the reason that caused the difference of vibrating hammer times for different samples.

The difference of the vibrating hammer times is obtained in relation to the maximum dry density obtained in the modified proctor. The vibrating hammer time is calculated with 97-98% of the density of the modified proctor.

In section 3.4, how does the CRT affected the mechanical properties of the designed mixtures?

Thank you very much for your comment, the document has been completed with (line 252):

“Particularly, due to the morphology and low water absorption shown by the CRT, the interaction between the particles and the cement does not materialize as with a natural or recycled aggregate, obtaining a loss of resistance in the mixtures that incorporate CRT”

Line 282-284, please discuss the possible reason for the neutralization effect of Mixed Recycled Aggregates in the article.

We are grateful by the opinion. Physical parameters and diffusion allowed estimating the relevance of porosity, density and absorption of MRA on diffusion release of the metals in study.

In Table 3, what is RAA stand for? In Table 4, what are RMA and RAB stand for?

We completely agree with your suggestion. After reviewing the Tables, we have observed a typographic error. The error has been corrected.

Reviewer 4 Report

The authors present a research work related to the “Feasible use of cathode ray tube glass (CRT) and recycled aggregates as unbound and cement-treated granular materials for road subbases”, in which the feasibility of incorporating CRT glass waste with recycled aggregates (RA) for civil engineering applications were studied.

In the abstract, please be more specific with the area of application. How does the CRT glass influence the mixtures durability? Can the composition of cathode ray tube glass vary from producer to producer? How does this aspect influence the recycling process? Rewrite properly: Line 64: n cement mortar Line 70: However, RA, particularly MRA, the sulphate content of the recycled aggregates (mostly gypsum) is one of the properties of the recycling itself. Line 250: 7d Line 85: Table 1. Measurement units. Line 125: Table 3. Measurement units. Insert references for standards. Future research directions may also be mentioned.

Author Response

The authors present a research work related to the “Feasible use of cathode ray tube glass (CRT) and recycled aggregates as unbound and cement-treated granular materials for road subbases”, in which the feasibility of incorporating CRT glass waste with recycled aggregates (RA) for civil engineering applications were studied.

In the abstract, please be more specific with the area of application.

Thank you very much for your suggestion, the application area has been included in the abstract .

How does the CRT glass influence the mixtures durability?

We are working on a research on the durability of mixtures with CRT glass, I can tell you that the use of CRT does not affect durability, a natural aggregate behaves similarly

Can the composition of cathode ray tube glass vary from producer to producer? How does this aspect influence the recycling process?

According to Méar et al. 2006, currently, collected monitors are dismantled and treated, and the CRT glass generally ends up in a special landfill licensed for hazardous waste. Waste CRT glass can be classified as being part of either color or black & white monitors, and by their manufacturer.it is possible to distinguish three main categories of cathode-ray tubes:

black & white glass: “barium” containing glass;

color panel glass: “barium – strontium” containing glass;

color funnel glass: "lead" containing glass.

In this paper, the crt studied was mainly of televisions or color monitors.

Méar, F., Yot, P., Cambon, M., & Ribes, M. (2006). The characterization of waste cathode-ray tube glass. Waste management, 26 (12), 1468-1476

Line 64: n cement mortar Line 70: However, RA, particularly MRA, the sulphate content of the recycled aggregates (mostly gypsum) is one of the properties of the recycling itself. Line 250: 7dLine 85: Table 1. Measurement units. Line 125: Table 3. Measurement units. Insert references for standards. Future research directions may also be mentioned. 

We are grateful by the opinion. It has been changed

Round 2

Reviewer 3 Report

The manuscript has been modified and authors have addressed the reviewer's comments.